# Parenting Record Handbook: The Needs of Mothers Raising Low Birth Weight Infants

**DOI:** 10.3390/ijerph19052520

**Published:** 2022-02-22

**Authors:** Yukiko Tomoyasu, Ikuko Sobue, Md Moshiur Rahman

**Affiliations:** 1Faculty of Nursing, Hiroshima International University, Kure 737-0112, Japan; 2Graduate School of Biomedical and Health Sciences, Hiroshima University, Hiroshima 734-8553, Japan; moshiur@hiroshima-u.ac.jp

**Keywords:** low birth weight infant, mother, postdischarge, growth and development evaluation, qualitative research

## Abstract

This study investigated the necessity for a parenting record handbook that is specifically tailored to the needs of low birth weight infants (LBWIs) and their families, especially mothers, who face parenting difficulties and challenges. The participants were 20 mothers, raising LBWIs, discharged from the neonatal intensive care unit. The mean age and weight of the children were 2.75 ± 0.35 years and 1417.50 ± 152.06 g, respectively; the mean duration of neonatal intensive care unit hospitalization was 78.75 ± 14.10 days. At the time of the study, 35% (7/20) were nursery children, 10% (2/20) were kindergarten children, 20% (4/20) were using rehabilitation centers, and 10% (2/20) were using the medical rehabilitation handbook. The needs of the mothers were investigated through focus group interviews or individual interviews, and content analyses were performed. The mothers required the promotion of peer support that assists the alleviation of mental burden and postpartum mental and physical care, as well as the publication of counseling service counters and reliable information sources for parenting difficulties in the parenting record handbook. The mothers required the publication and recording of the growth indicators of LBWIs, parenting records, information management of children since birth, and for the handbook to function as a multidisciplinary information sharing tool. In addition, the requirements for the parenting record handbook were the early provision of the parenting record handbook and measures to cope with poor maternal physical condition. The results of this study suggest that mothers with LBWIs require a parenting record handbook that can provide comprehensive maternal and child health assurance, starting from pregnancy, to resolve childcare difficulties for LBWIs, as well as mental support.

## 1. Introduction

Every year, approximately 15 million infants worldwide are born prematurely (before 37 weeks of pregnancy) [1,2]. However, the survival rate of premature infants has improved dramatically, not only in Japan, but also in other countries, due to advancements in perinatal care and improvements in the medical system [3]. Despite the improvements in survival rate, the number of children with neurological sequelae, such as cerebral palsy and mental retardation, as well as children requiring continuous medical care, is increasing [4,5,6,7,8,9]. Even in cases without neurological impairment, the development of extremely low birth weight infants (LBWIs) is slower than that of the average of healthy infants, and the shorter the gestational age, the lower the growth catch-up rate [10,11]. LBWIs experience motor developmental delay from infancy [12,13] and poor adaptation of group life from early childhood [14]. Many LBWIs face difficulties in learning, behavior, and social development after entering school [15,16]. LBWIs are at increased risk for attention-deficit hyperactivity disorder and autism spectrum disorders [17].

Thus, mothers face difficulties in appropriately responding to the children’s various reactions, and their negative emotions toward their children become stronger. Mothers of LBWIs have a higher risk of parenting anxiety (mental suffering) and depression than mothers of full-term infants [18,19,20].

The maternal and child health handbook enables the unified management of information, regarding maternal and child health from pregnancy, delivery to infancy; thus, it is a continuous assurance tool of multiprofessional maternal and child health services in various timings and instances. The maternal and child health handbook consists of development evaluation and parenting information for full-term infants, and the entry fields are for full-term infants. Thus, the maternal and child health handbook often does not match the growth evaluation and parenting of LBWIs, and they cannot fully guarantee the support for LBWIs and their families.

Therefore, some self-governing bodies and medical institutions have created and used a parenting record handbook for LBWIs. The Little Baby Handbook (LBH) includes information on the growth and development of LBWIs, raising and care policies, and messages for mothers and families of LBWIs from mothers with similar experiences [21]. Due to the lack of support of public services for the home care of LBWIs, it is expected that the LBH will play an important role in filling this gap. However, to the best of our knowledge, no survey had been performed to evaluate the utility of the LBH, except the research investigated in the focus group interview for mothers of LBWIs [21].

The mothers of LBWIs who used the LBH experienced a sense of pleasure from recording the growth and development of their children, and they acquired prospects of their development through messages from mothers with similar experiences. They evaluated the LBH as a useful tool for assessing the growth and development of their children and beneficial source for their health and medical care information. The results of our previous study suggest that the LBH empowers the mothers of LBWIs to change their negative feelings and behaviors to positive ones, thereby enhancing their quality of life. However, because the LBH user survey focused on the usefulness of LBH, it is possible that other important problems or needs of mothers of LBWIs, not listed in the LBH, have not been fully investigated.

To further improve the LBH, it is necessary to determine the content that needs to be added to the LBH, based on the parenting challenges and problems of mothers not using the LBH. The added contents can be expected to contribute new information and perspectives to the LBH. Based on the parenting difficulties and troubles of mothers of LBWIs, who are not using the LBH, this study investigated the necessary contents for the parenting record handbook, in order to create one that matches the needs of LBWIs and their mothers and families.

## 2. Materials and Methods

### 2.1. Research Design

This study was a qualitative inductive research.

### 2.2. Participants and Selection

The participants were the mothers of LBWIs (birth weight < 2500 g), at the age of the utilization of the maternal and child health handbook (0 years old to before entering elementary school). The exclusion criterion was mothers with a strong tendency for postpartum depression.

### 2.3. Procedure

The representative of an association of parents of LBWIs in prefecture A was requested to recommend participant candidates, based on the participant selection and exclusion criteria. Research descriptions and interview guides were sent to the participant candidates recommended by the representative. The researchers explained the study’s aims to the participants who agreed to participate in the research, obtained their written consent to participate and cooperate for this study (and to present them at academic conferences or submit to academic journals after protecting individual anonymity), and conducted interviews.

To prevent COVID-19 infection, focus group interviews were conducted using an online meeting platform (Zoom). When there were difficulties in the scheduling of focus group interviews, individual interviews were conducted by telephone. The interview content included challenges with parenting after the neonatal intensive care unit discharge and coping methods, desired support, and requested content for the parenting record handbook for LBWIs. With the participants’ consent, the focus group interviews were video recorded and individual interviews were audio recorded. The participants’ age and employment status, family structure, condition of LBWIs upon birth, and current utilization of medical and welfare services were surveyed by mail.

### 2.4. Analysis Methods

The analysis was supervised by researchers specialized in qualitative research. Without using qualitative data analysis software, transcripts were created from the recorded data of focus group and telephone interviews, and content analyses were performed. “Important items” were extracted from the participants’ “raw” expressions, and the meanings were probed and categorized [22]. To obtain a more detailed understanding of the concerns of mothers with low birth weight infants, their childcare difficulties and support needs were analyzed. We focused on their needs for the parenting record handbook, based on the difficulties associated with caring for a low birth weight infant. Therefore, we organized 38 categories into 24 on maternal difficulties and 14 on maternal support needs and extracted 8 core categories on needs for the parenting record handbook. Core categories were placed in chronological order because the difficulties and needs of participants spanned from pregnancy to the present (interview).

### 2.5. Ethical Considerations

Interview guides were added to the explanatory documents during recruitment, so that the participants could understand the interview contents and participate without anxiety. The participants were informed that participation was voluntary, and they could withdraw at any time. Measures were taken to ensure the anonymity and privacy of participants in interviews. This research was conducted with the approval of the Ethics Committee for Epidemiology of Hiroshima University (approval number: number: E-1879-1) and Epidemiological Research Ethics Review Committee of Hiroshima International University (approval number: Rin 20-014).

## 3. Results

### 3.1. Demographic Variables

The research participants were 20 mothers, raising LBWIs after NICU discharge (Table 1). Two mothers with school children were added as the participants of this study, due to their strong desire to participate. Moreover, 60% of the mothers were in their 30s, 95% had a nuclear family, and 60% were employed. The mean age of the LBWIs was 2.75 ± 0.35 years, with a mean birth weight of 1417.50 ± 152.06 g (40% had a birth weight of <1000 g) (Table 2). The mean gestational age at delivery was 31.25 ± 1.06 weeks, and the mean duration of NICU hospitalization was 78.75 ± 14.10 days. At the time of the study, 35% (7/20) were nursery children, 10% (2/20) were kindergarten children, 20% (4/20) were using rehabilitation centers, and 10% (2/20) were using the medical rehabilitation handbook.

### 3.2. Analysis Results of Focus Group Interviews and Individual Interviews

The focus group interviews included 14 participants and were conducted with 4–5 participants, for an average of 68 min, whereas the telephone interviews involved six participants, for an average of 29 min per participant. There were 77 subcategories, 38 categories, and 8 core categories that were extracted. The core categories are shown with [], categories with « », subcategories with < >, and contents narrated by the research participants in interviews, with “ ” words, in ( ), were added by the researchers to supplement the situations.

The growth situation of the LBWIs, as well as the mothers’ sense of difficulty, comprised of the process from pregnancy, birth of LBWIs, NICU hospitalization, and discharge, until present. The contents required in the parenting record handbook consisted of supporting needs, based on the contents of this process. The mothers’ needs for the parenting record handbook were the [promotion of peer support] that assists the [alleviation of mental burden] and [postpartum mental and physical care], as well as [counseling service counters and reliable information source for parenting difficulties]. In addition, they pointed out the importance of [growth indicators of LBWIs], [parenting records], and [information management of children since birth] in the parenting record handbook, regarding the current and future health of LBWIs, and required it to function as [a multidisciplinary information sharing tool].

#### 3.2.1. Required Contents in the Parenting Record Handbook: Alleviation of Mental Burden and Promotion of Peer Support

The mothers faced the «shock of premature delivery» and «confusion and self-condemnation for the birth of LBWIs», and their infants’ NICU hospitalization induced «mother–infant separation anxiety». The mothers narrated that the «lack of information regarding premature delivery», «lack of opportunities to learn about LBWIs», and «lack of peer support» during pregnancy increased the shock and confusion of premature delivery. In addition, the mothers spoke of the importance of mental support through peer support from the experience of obtaining sympathy after «consulting mothers with the same experience», as well as the experience of joining «LBWI clubs» via parenting.

The mothers anticipated the [alleviation of mental burden] from the parenting record handbook. The mothers expressed the need for the [promotion of peer support], through messages and experience-sharing of mothers with the same experience, as well as the publication of information about «LBWI clubs» in the parenting record handbook. The mothers perceived that «words and experience-sharing of more experienced mothers in the handbook for LBWIs are encouraging». In addition, the mothers perceived that the «early provision of the parenting record handbook for LBWIs» led to the acquisition of information regarding premature delivery and «alleviated the shock when facing the birth of LBWIs».

“I am losing my head due to the sudden delivery, so if such experiences are written, I know I am not alone and can feel reassured”.

#### 3.2.2. Required Contents in the Parenting Record Handbook: Postpartum Mental and Physical Care

The mothers experienced «poor maternal physical condition», due to premature delivery, long-term hospitalization, postdischarge «LBWI (specific) poor physical condition», and «parenting difficulties», such as <lack of breast milk> and <not eating meals (weaning food)>. They told us that, despite the «mental and physical overload due to parenting», they <put off their own matters>, but they know that <maternal health is important>. Furthermore, they pointed out that there was «no support for mothers» at the hospital during delivery. For the need of [postpartum mental and physical care], the mothers required the publication of contents, regarding «the importance of postpartum physical care», in the parenting record handbook.

“When I felt frustrated and ate only snacks or bread, it got worse rapidly, causing poor physical condition or depression, so it would be good to have a page for mothers’ care, or a page for nutrition or self-management”.

#### 3.2.3. Required Contents in the Parenting Record Handbook: Counseling Service Counters and Reliable Information Sources for Parenting Difficulties, Multidisciplinary Information Sharing Tool

The «mothers’ difficulties in obtaining information» regarding LBWIs occurred because of the «lack of opportunities to learn about LBWIs» due to «poor maternal physical condition». In addition, regarding the «pain of lack of information» regarding «LBWI (specific) poor physical condition» and «parenting difficulties», the mothers said that <books and the internet are biased toward information about full-term infants>, and <no matter how you search, there is no information leading to parents’ peace of mind> about the illnesses and parenting information of LBWIs. The mothers desired the publication of «expert knowledge about LBWI-specific symptoms», «reliable information by the government and experts», and «counseling service counters for LBWIs» in the parenting record handbook.

In addition, the mothers pointed out the «lack of cooperation of public institutions» about «developmental disorders» and «developmental support» of LBWIs. The mothers pointed out that they <do not know the counseling service counters>, <the person-in-charge changes every time>, and <there is no leader in cooperation>, regarding the support for their children’s development and school entry. Furthermore, they said that they had <no choice but to act on their own>, when dealing with «measures to support growth». The mothers expressed the need for [a multidisciplinary information sharing tool] that enables «information sharing between professionals and families» about LBWIs in the parenting record handbook.

“If there is information such as common illnesses (among low birth weight infants), I can know that it is normal for babies born tiny to develop them, and feel relieved”.

“If everything about physiotherapists is written, and everything about babies born early is incorporated in that one book, I can bring it and go to rehabilitation centers or get developmental support”.

#### 3.2.4. Required Contents in the Parenting Record Handbook: Growth Indicators of LBWIs, Parenting Records, and Information Management of Children since Birth

For the mothers, the LBWIs’ current «growth and developmental delay» caused «anxiety about growth and developmental delay». Knowing that their children’s development is delayed «hurts due to comparison with the other children (full-term infants)», and the mothers expressed that they <know about growth and developmental delay from the comparison with healthy children>, <dislike talking about growth during medical examinations>, and feel the <hurt that things cannot be changed>. In addition, they pointed out that the «maternal and child health handbook unsuitable for LBWIs», and they <cannot use the growth curve of healthy children> and <cannot record development even with corrected age>. The mothers also said that they could not keep a growth record of their children, as the current maternal and child health handbook only deals with the growth of full-term infants.

Thus, the mothers said that they «want a growth curve that can record growth from birth» and desired to be «able to record the children’s growth at their pace» in the parenting record handbook. The mothers perceived <knowing the slightly advanced growth of children with the same experience> as «obtaining a future outlook about growth» and required the publication about the growth of LBWIs through more experienced mothers in the parenting record handbook.

“(The maternal and child health handbook) was totally blank and nothing could be written”.

”It would be helpful if there is something like a standard from more experienced mothers, such as mothers of children who are slightly older than my own child telling, ‘My child was like this at two years old, but now he/she is like this”.

## 4. Discussion

The mothers were shocked by the birth of LBWIs, suffered from self-condemnation for giving birth to LBWIs, and were troubled by the lack of opportunities for peer support and to learn about LBWIs. The mothers required the [promotion of peer support], which assists in the [alleviation of mental burden] and [postpartum mental and physical care], as well as [counseling service counters and reliable information source for parenting difficulties], in the parenting record handbook.

It was a new finding that the mothers needed to alleviate their mental burden through the early provision of the parenting record handbook since pregnancy. For mothers, a premature delivery is traumatic [23]. The postpsychological traumatic stress reaction of the mothers of premature infants is significantly higher than that of the mothers of full-term infants, from immediately after delivery to 14 months after delivery [24]. The risk of postpartum depression of the mothers of LBWIs is 2–3 times that of the mothers of full-term infants [25]. In the case of nondistressed mothers, very LBWIs showed higher mean scores of hearing and language in child development areas than full-term infants [26]. This relationship does not hold true in the case of LBWIs with distressed mothers. The protective effects of nondistress of a mother and potential negative effects of a mother’s distress on infant development would support the specific interventions to reduce mother’s distress.

Although poor social support is an important risk factor for the development of postpartum posttraumatic stress symptoms in parents with very LBWIs [27], early intervention for the mental stress of mothers of LBWIs mitigates their postpsychological traumatic stress reactions [28]. The social support received by the mother, from the time of the infant’s premature birth, seems to inhibit the development of depressive symptoms [29]. The messages of mothers with the same experiences published in the LBH have rescued, supported, and encouraged the mothers [21]. The participation in the parents’ associations of LBWIs, published in the LBH, connects mothers who understand one another’s experiences, and they become a source of continuous support for the mothers. These findings support the need for the early provision of the parenting record handbook to mothers of LBWIs.

Another novel finding of this study was that poor maternal physical condition, due to premature delivery, long-term hospitalization, and mental and physical overload, due to parenting, caused the mothers’ difficulties in obtaining information. As the books and information on the internet are biased toward full-term infants, these mothers had a lack of information about «LBWI (specific) poor physical condition» and «parenting difficulties». There are concerns about the reliability of information from online resources [30], and there is no guarantee in the quality of online information [31,32]. The parents of LBWIs struggle with their children’s development, care, and acquiring supporting information [31]. Conversely, the characteristics of LBWIs published in the LBH (risk of complications and growth), as well as the publication of public subsidies and counseling institutions, promote the mothers’ knowledge and information utilization and improve maternal and child quality of life [21]. The consistency between the research participants’ requirements for the parenting record handbook and contents of the LBH, as well as the presentation of new contents (early distribution of the parenting record handbook and measures to cope with poor maternal physical condition), suggest improvements in the contents and utilization of the parenting record handbook.

The mothers pointed out the importance of [growth indicators of LBWIs], [parenting records], and [information management of children since birth] in the parenting record handbook as the current maternal and child health handbook is unsuitable for the growth management of LBWIs, and there is a higher chance that LBWIs will require medical and childcare support. In addition, they required the parenting record handbook to function as [a multidisciplinary information sharing tool].

Even in the 12 corrected months, the growth of LBWIs is often less than the growth values equivalent to their corrected age [10]. As the maternal and child health handbook in Japan deals with the growth and development of full-term infants, the parents of extremely LBWIs cannot record their children’s growth in the maternal and child health handbook, which results in the mothers not knowing the standard of their children’s growth and development [33]. In the LBH, the growth values of LBWIs can be written in the LBWI growth curve. In addition, in the list of developments, “holding up the head” and “walking without support” can be written in the corrected age and age, in months, (age in year) that their children could achieve them. The users commented that they could check the growth process of LBWIs with the LBH and enjoy their children’s growth at their pace [21].

The research participants pointed out the importance of information management of children since birth. They felt it difficult to explain the progress of children after birth and desired cure/care briefing from medical professionals in the parenting record handbook.

Many LWBIs have an increased risk of complications, such as cerebral palsy [4,5,6,7,8,9], respiratory diseases [34], and health issues. The parenting record handbook, which is an information sharing tool between multiple professions and families, is important in the developmental support for children who require early rehabilitation and continuous medical care. The users of the LBH also have issues with the LBH, as a multidisciplinary cooperation tool, and the statements of medical institutions and professionals are desired [21].

The research participants’ requirements for the parenting record handbook are contents that add information of children since birth, and it must function as a tool to share multidisciplinary information, regarding the growth indicators of LBWIs, in the LBH. The mothers of low LBWIs also require a tool that can provide continuous assurance and information on maternal and child health services in the parenting record handbook.

### Limitations and Further Research

First, this was a qualitative survey of mothers raising LBWIs, with the participants belonging to one “parents’ association” in one region. In the future, to generalize the needs for the parenting record handbook, a quantitative survey with a larger number of participants is necessary. Second, the participants of this research were mothers of children who did not require medical care. To create a parenting record handbook for LBWIs, a needs analysis of mothers raising children under medical care, and based on the gestational week of LBWIs, are necessary. Third, since only mothers were surveyed in the present study, it is necessary to interview the fathers of low birth weight infants. A more detailed understanding of the struggles and needs of fathers will help to develop the handbook into more family-centered content. A low birth weight infant and family-centered handbook will be useful for supporting diverse families. Last, as the parenting record handbook is an information sharing tool between families and multiple professions, the opinions of professionals from medical institutions and administrative institutions (about the parenting record handbook) are necessary.

## 5. Conclusions

The mothers required the promotion of peer support, which alleviates mental burden and assists in their postpartum mental and physical care, as well as the publication of counseling service counters and reliable information sources to overcome parenting difficulties, in the parenting record handbook. The mothers required the parenting record handbook to include growth indicators of LBWIs, parenting records, management information of children since birth, and for it to function as a multidisciplinary information sharing tool. The mothers’ requirements for the parenting record handbook were the early provision of the parenting record handbook and measures to cope with poor maternal physical condition.

## Figures and Tables

**Table 1 ijerph-19-02520-t001:** Demographic of study participant’s family (*n* = 20).

Variable	Item	*n*	%
		Mean	SD
Maternal age	20s	2	10
	30s	12	60
	40s	6	30
Employment	Full-time employee	7	35
	Part-timer	2	10
	On leave	2	10
	Self-employed	1	5
	No job	8	40
Sibling number	0	8	40
	1	7	35
	2	4	20
	3	1	5
Sibling age		6.89	0.90

SD: standard deviation.

**Table 2 ijerph-19-02520-t002:** Demographic of low birth weight infant (*n* = 20).

Variable	Unit	Mean	SD
Age	Years	2.75	0.35
Week of delivery	Week	31.25	1.06
Birth weight	Gram	1417.50	152.06
NICU hospitalization on period	Day	78.75	14.10
	Item	*n*	%
Sex	Female	9	45
	Male	11	55
Age	One-year	3	15
	Two-year	10	50
	Three-year	1	5
	Four-year	3	15
	Five-year	1	5
	Six-year	2	10
Birth weight	1500 g =<, <2500 g	9	45
	1000 g =<, <1500 g	3	15
	<1000 g	8	40
Week of Delivery	=<37 weeks	3	15
	34 weeks =<, <37 weeks	5	25
	<34 weeks	12	60
NICU hospitalization on period	<one month	5	25
	1 month =<, 2 months	3	15
	2 months =<, 3 months	3	15
	3 months =<, <4 months	4	20
	4 months =<	5	25
Medical/welfare service	Nursery	7	35
	Kindergarten	2	10
	Rehabilitation center	4 *^,^^#^	20
	After-school day service	1 *	5
	Special support school	1 *	5
	Counseling support for children with disabilities	1 *	5
	Outpatient rehabilitation	1 ^#^	5
	Medical rehabilitation handbook	2	10

SD: standard deviation, *^,#^: same child.

## Data Availability

The data presented in this study are available on request from the corresponding author. The data are not publicly available due to research participants’ privacy.

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
