# Peer review of "Parenting Record Handbook: The Needs of Mothers Raising Low Birth Weight Infants"

_ijerph, 2022, doi:10.3390/ijerph19052520_

Round 1

Reviewer 1 Report

The review of an article titled "Parenting record handbook: The needs of mothers raising low- birth-weight infants „

This article raises a very important issue about problems for parents of children born with low birth weight infants.

This project is qualitative inductive research. This information is very subjective.

Author/s show in tables very important information about maternal age, birth weight, week of delivery, etc., but the most important information is too general.

Author/s should show us which of the pieces of information are more or less important ?? Too many categories and too many subcategories.

Comments:

Table 1., table2.:

sibling age- nil: replace ,,-''

Medical/Welfare Service nil: replace,,-''

Employment nil: replace,,-''

The Parenting record handbook – Author/s recruit only mothers

Author Response

Reviewer 1:

Comments and Suggestions for Authors:

The review of an article titled "Parenting record handbook: The needs of mothers raising low- birth-weight infants „ This article raises a very important issue about problems for parents of children born with low birth weight infants. This project is qualitative inductive research. This information is very subjective.

Response: We wish to thank the Reviewer for these useful comments, which have improved the quality of our manuscript.

Comment: Author/s show in tables very important information about maternal age, birth weight, week of delivery, etc., but the most important information is too general.

Response: Thank you for pointing this important observation. We have summarized the demographic variables of the participants and low-birth-weight infants as basic statistics and revised tables 1 and 2, and focused on important information.    

Comment: Author/s should show us which of the pieces of information are more or less important ?? Too many categories and too many subcategories.

Response: Thank you for your precious comment. We agree that our explanation of the process of extracting subcategories and categories from the data was insufficient. We analyzed data on the childcare difficulties and support needs of mothers with low-birth-weight infants. It was important to obtain a more detailed understanding of the concerns of these mothers. We focused on their comments on modifications to the parenting record handbook to reflect the difficulties associated with caring for a low-birth-weight infant. This led us to organize 38 categories into 24 on maternal difficulties and 14 categories on maternal needs, and we extracted 8 core categories on needs for the parenting record handbook. The difficulties and needs of mothers spanned from pregnancy to the present. The number of categories was large due to analyses of the difficulties and needs encountered at each stage. Based on the Reviewer’s comment, we have revised and rewrote the text in the ‘Analysis Methods’ section.

Comment:

Table 1., table2.:

sibling age- nil: replace ,,-''  

Medical/Welfare Service nil: replace,,-''

Employment nil: replace,,-''

Response: We have revised these expressions. We summarized demographic variables for the participants and low-birth-weight infants as basic statistics and revised the tables. The term ‘nil’ for sibling age was deleted, and sibling numbers and the mean and SD of sibling ages were described. The term ‘nil’ for medical/welfare services was deleted, and ‘Nil’ for employment was replaced with ‘no job’. We replaced ‘55% had a birth weight of <1,500 g’ with ‘40% had a birth weight of <1,000 g.’ in the paragraph on demographic variables.

Comment: The Parenting record handbook – Author/s recruit only mothers

Response: Thank you for the Reviewer’s important remarks. The parenting record handbook aims to provide low-birth-weight infant and family-centered care. Nowadays, families are becoming more diverse including same-sex marriages and single mothers (fathers). Therefore, we aim to create a parenting record handbook that provides support for all families. To further improve the handbook, we would like to consider the content of the handbook with the distress and support needs of mothers who have given birth to low birth weight infants at its core. We also plan to interview the fathers of low-birth-weight infants about their concerns and needs. The findings of interviews with fathers will enrich the content of the handbook from mother support to family support. Based on the Reviewer's suggestion, we added the text in the ‘Limitations and Further Research’ section as “Since only mothers were surveyed in the present study, it is necessary to interview the fathers of low-birth-weight infants. A more detailed understanding of the struggles and needs of fathers will help to develop the handbook into that with a more family-centered content. A low-birth-weight infant and family-centered handbook will be useful for supporting diverse families.”

Reviewer 2 Report

It is an interesting topic although similar studies have been published. I would suggest, that you provide some more information regarding your results analysis.

To be more specific I would recommend, you try to combine responses in a crosstab in an attempt to identify if there were any kind of groups showing, e.g. combining similarities in problem reporting with level of perceived usefuleness of the handbook.

It would also be very useful in terms of your study generability if you could analyse a larger sample than that of the 20 cases.

Author Response

Reviewer 2:

Comments and Suggestions for Authors:

It is an interesting topic although similar studies have been published.

Comment: I would suggest, that you provide some more information regarding your results analysis.

To be more specific I would recommend, you try to combine responses in a crosstab in an attempt to identify if there were any kind of groups showing, e.g. combining similarities in problem reporting with level of perceived usefuleness of the handbook.

Response: We thank the Reviewer for this important comment. We identified 24 categories on maternal difficulties and 14 on maternal support needs, and extracted 8 core categories on needs for the handbook. However, the characteristics of difficulties and support for each group were unclear. Since all participants were analyzed as one group, it was challenging to analyze the specific difficulties and needs of a particular group using this method of analysis. According to the suggestion provided, we identified nine infants who weighed >1500 g and eight who weighed <1,000 g; seven attended nursery school, two attended kindergarten, and only four needed rehabilitation. Although this information was not included in the manuscript, the mothers of several children weighing >1,500 grams stated that their children did not receive adequate support. We intend to group target children based on the relationship between birth weight and the current health status, and then compare the concerns and needs of their mothers. We sincerely appreciate the keen insight of the Reviewer regarding the significance of the handbook because we believe that the prediction of a healthy upbringing by health care providers does not mean that care is not necessary. We will continue our work in consideration of the results of this new analysis, which offers important information on how to provide the best support for low-birth-weight infants and their families who have not received sufficient care. We strongly appreciate the Reviewer's comment on this point.

Comment: It would also be very useful in terms of your study generability if you could analyse a larger sample than that of the 20 cases.

Response: Thank you for your helpful comments. A quantitative survey of a large sample is needed to improve the quality of the parenting record handbook and its usefulness for the families of low-birth-weight infants. We are preparing a questionnaire for the mothers of low-birth-weight infants who have not used this handbook. We have added the text in the ‘Limitations and Further Research’ section as “In the future, a quantitative survey of a larger number of participants is required to generalize needs for the parenting record handbook.”

Reviewer 3 Report

In this study authors investigated the necessity for a parenting record handbook that is specifically tailored to the needs of low-birth-weight infants (LBWIs) and their families, especially mothers, who face parenting difficulties and challenges and their results suggest that mothers with LBWIs require a parenting record handbook that can provide comprehensive maternal and child health assurance starting from pregnancy to resolve childcare difficulties for LBWIs as well as mental support.

It is well designed study and although the small number of subject included their results are important in order to provide integrated support for families of VLBWI and especially the mothers.

Author Response

Reviewer 3:

Comments and Suggestions for Authors:

In this study authors investigated the necessity for a parenting record handbook that is specifically tailored to the needs of low-birth-weight infants (LBWIs) and their families, especially mothers, who face parenting difficulties and challenges and their results suggest that mothers with LBWIs require a parenting record handbook that can provide comprehensive maternal and child health assurance starting from pregnancy to resolve childcare difficulties for LBWIs as well as mental support.

It is well designed study and although the small number of subject included their results are important in order to provide integrated support for families of VLBWI and especially the mothers.

Response: We wish to express our appreciation to the Reviewer for the deep understanding and insightful comments on the issue of mothers with low-birth-weight infants. We are encouraged by these comments and will perform further research on this issue.